# Evidence Gaps and Lessons in the Early Detection of Atrial Fibrillation: A Prospective Study in a Primary Care Setting (PREFATE Study) [note 1]

**DOI:** 10.3390/biomedicines13010119

**Published:** 2025-01-07

**Authors:** Josep L. Clua-Espuny, Alba Hernández-Pinilla, Delicia Gentille-Lorente, Eulàlia Muria-Subirats, Teresa Forcadell-Arenas, Cinta de Diego-Cabanes, Domingo Ribas-Seguí, Anna Diaz-Vilarasau, Cristina Molins-Rojas, Meritxell Palleja-Millan, Eva M. Satué-Gracia, Francisco Martín-Luján

**Affiliations:** 1Ebrictus Research Group, Research Support Unit Terres de l’Ebre, Institut Universitari d’Investigació en Atenció Primària Jordi Gol (IDIAP Jordi Gol), 43500 Tortosa, Spain; 2Primary Health-Care Centre Tortosa Est, Institut Català de la Salut, Primary Care Service (SAP) Terres de l’Ebre, 43500 Tortosa, Spain; 3Servicio de Atención Primaria Camp de Tarragona, Institut Català de la Salut, 43761 Tarragona, Spain; ahernandez.tgn.ics@gencat.cat; 4Servicio de Cardiología, Hospital Virgen de la Cinta de Tortosa, Institut Català de la Salut, 43500 Tortosa, Spain; dgentille.ebre.ics@gencat.cat; 5Primary Health-Care Centre Amposta, Institute Català de la Salut, Primary Care Service (SAP) Terres de l’Ebre, 43870 Amposta, Spain; eumuria@gmail.com; 6Primary Health-Care Centre Tortosa Oest, Institute Català de la Salut, Primary Care Service (SAP) Terres de l’Ebre, 43500 Tortosa, Spain; tforcadella.ebre.ics@gencat.cat; 7Primary Health-Care Centre Salou, Institute Català de la Salut, Department of Primary Care Camp de Tarragona, 43840 Salou, Spain; mcdiego.tgn.ics@gencat.cat; 8Department of Primary Care Camp de Tarragona, Institut Català de la Salut, 43005 Tarragona, Spain; dribas.hj23.ics@gencat.cat (D.R.-S.); anndiaz.tgn.ics@gencat.cat (A.D.-V.); 9Primary Health-Care Centre Sant Pere I Sant Pau, Institute Català de la Salut, Department of Primary Care Camp de Tarragona, 43007 Tarragona, Spain; cmolins.tgn.ics@gencat.cat; 10Unitat de Suport a la Recerca Camp de Tarragona-Reus, Institut de Recerca en Atenció Primària Jordi Gol, 43201 Reus, Spain; mpalleja@idiapjgol.info; 11Unitat de Suport a la Recerca, Fundació Institut Universitari per a la Recerca a l’Atenció Primària de Salut Jordi Gol i Gurina (IDIAPJGol), 43201 Reus, Spain; esatue.tgn.ics@gencat.cat (E.M.S.-G.); fmartin.tgn.ics@gencat.cat (F.M.-L.); 12Departament de Medicina i Ciències de la Salut, Universitat Rovira i Virgili, 43206 Reus, Spain

**Keywords:** arrhythmias, atrial fibrillation, cardiac/diagnosis, heart rate determination, echocardiography/statistics and numerical data, electrocardiography, ambulatory/standards, diagnostic techniques and procedures, clinical risk scores, device detected atrial fibrillation, ischemic stroke

## Abstract

**Background/Objectives:** In Europe, the prevalence of AF is expected to increase 2.5-fold over the next 50 years with a lifetime risk of 1 in 3–5 individuals after the age of 55 years and a 34% rise in AF-related strokes. The PREFATE project investigates evidence gaps in the early detection of atrial fibrillation in high-risk populations within primary care. This study aims to estimate the prevalence of device-detected atrial fibrillation (DDAF) and assess the feasibility and impact of systematic screening in routine primary care. **Methods:** The prospective cohort study (NCT 05772806) included 149 patients aged 65–85 years, identified as high-risk for AF. Participants underwent 14 days of cardiac rhythm monitoring using the Fibricheck^®^ app (CE certificate number BE16/819942412), alongside evaluations with standard ECG and transthoracic echocardiography. The primary endpoint was a new AF diagnosis confirmed by ECG or Holter monitoring. Statistical analyses examined relationships between AF and clinical, echocardiographic, and biomarker variables. **Results:** A total of 18 cases (12.08%) were identified as positive for possible DDAF using FibriCheck^®^ and 13 new cases of AF were diagnosed during follow-up, with a 71.4-fold higher probability of confirming AF in FibriCheck^®^-positive individuals than in FibriCheck^®^-negative individuals, resulting in a post-test odds of 87.7%. Significant echocardiographic markers of AF included reduced left atrial strain (<26%) and left atrial ejection fraction (<50%). MVP ECG risk scores ≥ 4 strongly predicted new AF diagnoses. However, inconsistencies in monitoring outcomes and limitations in current guidelines, particularly regarding AF burden, were observed. **Conclusions:** The study underscores the feasibility and utility of AF screening in primary care but identifies critical gaps in diagnostic criteria, anticoagulation thresholds, and guideline recommendations.

## 1. Introduction

Atrial fibrillation (AF) and stroke are common conditions that disproportionately affect the elderly population. In Europe, the prevalence of AF is expected to increase 2.5-fold over the next 50 years [1,2] with a lifetime risk of 1 in 3–5 individuals after the age of 55 years (37% IC95% 34.7–39.6%) [3,4] and a 34% rise in AF-related strokes is projected in the coming decades. Moreover, the number of ischemic strokes in individuals over 80 is expected to triple between 2016 and 2060. Additionally, a 27% increase is anticipated among stroke survivors who develop AF or related conditions. Improving effective risk assessment for atrial fibrillation and its complications is key to primary and secondary prevention and will be essential to enhance stroke prevention and reduce the associated social and economic burden on the European population [5]. The Action Plan for Europe (2018–2030) emphasizes implementing detection and treatment programs in primary care to enhance the diagnosis and management of populations at risk of atrial fibrillation (AF) [1]. AF is a common condition in high-risk cardiovascular populations. Electrocardiogram (ECG) screening is crucial, along with echocardiography, to detect atrial cardiomyopathy—a structural and functional abnormality of the atrium that can serve as a substrate for AF and increase the risk of thromboembolic events, particularly stroke, even in the absence of arrhythmia. In patients with cardiac implantable electronic devices, the prevalence of asymptomatic device-detected AF episodes among individuals with risk factors but no prior AF diagnosis is approximately 30% [6,7], and they have introduced new definitions [8] such as clinically unrecognized AF, including device-detected AF (DDAF), sub-clinical AF (SCAF), atrial high-rate episodes (AHREs), frequent atrial premature complexes (APCs), or short runs (<30 s) of AF-like arrhythmias (micro-AF).

In the last 15 years, numerous clinical risk score models have been created to forecast the probability of developing AF over a 5–10-year period. These models have been experimentally used in clinical settings, revealing a surprisingly high occurrence of subclinical AF within the at-risk general population. Furthermore, these studies have demonstrated that screening for AF in high-risk individuals is both safe and advantageous.

However, there is a major gap in stroke prevention among undetected AF in the community [9]. Additionally, this arrhythmia is associated with increased risks of cognitive impairment, heart failure, sudden death, and cardiovascular morbidity [8,10,11]. Advances in screening technology have been made in tandem with the aging population and increasing atrial fibrillation prevalence. While several randomized controlled trials [12,13] demonstrate the efficacy of AF screening, it has not proven effective in reducing stroke incidence [8,14], and evidence indicates the need for using AF risk scores and new monitoring technologies in high-risk populations [15,16,17].

Physicians recognize that many high-risk patients may benefit from electronic devices capable of rhythm monitoring, which can reliably detect SCAF [4,18]. New technologies offer expanded screening opportunities. However, integrating these tools into clinical practice requires further high-quality evidence. Several uncertainties remain around the early diagnosis and management of AF. These include the lack of a clear definition for high-risk AF patients, appropriate diagnostic criteria for DDAF and SCAF, the contributions of ECG scoring and echocardiography, potential impacts on AF outcomes, the use of emerging treatments like ablation, and the criteria for thromboembolic risk scores related to anticoagulation and managing DDAF and SCAF in patients with complex morbidity burden remain critical issues.

In this paper, we present additional results from the PREFATE study [19]. The primary goal of the study was to estimate the prevalence of DDAF among patients identified as high-risk in primary care, exploring the increased risk associated with several markers of atrial dysfunction.

## 2. Materials and Methods

The PREFATE project is a prospective observational cohort study (1 January 2023–31 December 2024), with its protocol previously published and registered on ClinicalTrials.gov (NCT05772806). The study protocol and previous results have already been detailed in earlier publications [19,20,21]; therefore, they will not be repeated here.

The study protocol was reviewed and approved by the Independent Ethics Committee of the Foundation University Institute for Primary Health Care Research IDIAP Jordi Gol, expedient file 22/090-­P. All participants were properly informed about the study, and the consent form for participation was distributed and signed before being officially recruited.

The equipment was sourced by Qompium, Hasselt, Belgium (1 January 2023), which has developed a solution, FibriCheck (https://pages.fibricheck.com/portal-us/, accessed on 12 December 2024), an integrated mobile solution that allows for the swift and accurate detection of heart rhythm disorders. a screening and monitoring software tool to detect irregular heart rhythms for the area of diagnostics for heart rhythm disorders with a focus on Atrial Fibrillation. The certification of FibriCheck^®^ as a Medical Device was received on 28 June 2016 with certificate number BE16/819942412. FibriCheck^®^ has been successfully re-certified since 8 October 2019.

### 2.1. Participants

A randomized sample of patients was selected from the electronic health records of primary care centers managed by the Catalan Health Institute and located in Southern Catalonia (Spain). Figure 1 presents the Consolidated Standards of the Reporting Trials (CONSORT) flow diagram of the participants.

The inclusion criteria were as follows: (1) age between 65 and 85 years; (2) no prior diagnosis of AF (of any type), no history of stroke, and no current anticoagulant treatment; (3) high risk of AF [22]; (4) a CHA2DS2-VA score of ≥2; and (5) ability to use the Fibricheck^®^ App, either by the patient or their caregiver.

All participants were classified as being at high risk of developing AF based on the AF-risk calculator, which employs a previously validated predictive model for AF in the general population [22]. The follow-up period is set for two years, from 1 January 2023 to 31 December 2024.

### 2.2. Data Collection and Procedures

The primary outcome was the new diagnosis of AF during follow-up. Independent variables included those required for AF risk stratification, sociodemographic characteristics, clinical features, comorbidities, cardiovascular risk scores, biomarkers, and current cardiovascular treatments. Baseline data were collected from participants’ electronic clinical records and from tests and assessments performed on them, including ECG and echocardiogram. Echocardiogram data were reported by the cardiologist who conducted the test. Heart rate monitoring data from FibriCheck^®^ App were retrieved through the electronic device’s records [23].

Basal assessment and clinical follow-up included the evaluation of P-wave patterns on the ECG using the Morphology-Voltage-P-wave (MVP) ECG risk score [20] (Appendix A) and assessment of left atrial (LA) size and function via 2D transthoracic echocardiography (TTE). Additionally, cardiac rhythm monitoring over a 14-day period was performed using the FibriCheck^®^ App. All possible AF diagnoses were confirmed through ECG recording or Holter monitoring, with a minimum duration of 5 days [8,24]. A cardiologist reviewed and validated the AF diagnosis and the presence and severity of supraventricular arrhythmias. ETT was performed at the beginning of the study, Fibricheck^®^ monitoring on two occasions (with annual periodicity), and ECG every six months during the study follow-up.

#### 2.2.1. Electrocardiogram Study

A standard 12-lead ECG was performed for all patients, utilizing a filter setting of 150 Hz, a recording speed of 25 mm/s, and a calibration of 10 mm/mV. An experienced cardiologist analyzed all ECG recordings using a Zoom tool for detailed evaluation of MVP ECG score [25] (see Appendix A) and the presence of interatrial block (IAB) defined as prolonged P wave duration (≥120 ms) due to delayed transmission of the sinus impulse through the region of the Bachmann bundle [20].

#### 2.2.2. Echocardiogram Study

In accordance with the consensus document from the European Association of Cardiovascular Imaging/American Society of Echocardiography (EACVI/ASE) [26,27], a 2D transthoracic echocardiography was performed using the EPIQ 7 ultrasound system with an X5-1 transducer (Philips Medical Systems, Amsterdam, The Netherlands).

The examinations were conducted and analyzed by a single experienced cardiologist who was blinded to the patients’ medical histories and ECG parameters. All echocardiographic studies were digitally stored and subsequently analyzed offline using IntelliSpace Cardiovascular QLAB 15.0 software (IISCV-QLAB Philips Medical Systems, Amsterdam, The Netherlands). The left atrial indexed volume (LA-iV) was measured using the biplane disk summation method, normalized to the patient’s body surface area. The left atrial ejection fraction (LA-EF) was calculated using the Simpson method from both apical four- and two-chamber views. Additionally (Appendix A), the left atrial reservoir strain (LA-Sr) was determined and classified as ≤26% or >26%, based on evidence indicating that LA-Sr values ≤ 26% provide strong prognostic information regarding the risk of AF and/or ischemic stroke across various populations [28,29].

#### 2.2.3. External Monitoring Fibricheck^®^

The recordings generated by the FibriCheck^®^ App over a 14-day period, according to protocol [23], were reviewed by the principal investigator. A minimum of two daily recordings were required, although patients could perform as many as they deemed necessary. Each self-monitoring lasted 1 min. The total number of self-measurements and the percentage registered as probable AF were recorded.

The FibriCheck^®^ App provided an immediate report (<24 h) and a comprehensive final report at the end of the rhythm monitoring. Results were categorized as normal, atrial high-rate episodes (AHRE), atrial premature complexes (APCs), or flutter/AF. When the app indicated probable AF, confirmation via ECG was required. If ECG results were negative, patients were referred to cardiology for Holter monitoring in accordance with the European Heart Rhythm Association (EHRA) guidelines [30].

### 2.3. Sample Size

To detect a new diagnosis of AF with a difference of 0.1 units from the reference population (expected proportion in the reference group: 0.10) [22], a randomized sample size of 149 individuals was calculated. This estimation assumed a significance level (alpha risk) of 0.05, a statistical power (beta risk) of 80% with an alpha risk of 0.05, a beta risk of 0.20 for a two-sided test, and an anticipated loss to follow-up rate of less than 15%.

### 2.4. Statistical Analysis

Data were presented as frequencies and percentages for categorical variables and means and standard deviations for continuous variables, grouped by new AF status. To compare and assess possible differences between the two groups defined by new AF status, the chi-square test was used for categorical variables, while the Student’s *t*-test or Mann–Whitney U test was applied for continuous variables, depending on normality. The Shapiro–Wilk test was used to evaluate normality. Sensitivity and specificity along with predictive positive values and post-test odds were calculated for different suspicion diagnostic approaches to AF: Fibricheck^®^ detection, MVP ECG risk score > 4 or LA-Sr < 26%.

Analyses and data handling were performed using R Statistics software (R Foundation for Statistical Computing, Vienna, Austria; version 4.0.5).

## 3. Results

A total of 149 individuals aged > 65 years at high risk of AF were recruited. The average age of participants was 74.7 years [SD 5.11], and 64.4% were women.

Figure 1 presents the CONSORT flow diagram for participant recruitment and follow-up. It involved two rounds of cardiac rhythm monitoring using the FibriCheck^®^, baseline and follow-up ECGs, and Holter monitoring for diagnostic confirmation. Outcomes include AF detection, irregular rhythm findings, and non-AF diagnoses.

In total, 13 new AF cases were confirmed during the follow-up period among 149 high-risk individuals, corresponding to a number of needed-to-screen (NNS) of 12. Table 1 presents the baseline characteristics of the population, comparing the subgroup with a new diagnosis of AF to the subgroup without the diagnosis.

No significant differences were observed between the two groups. No significant differences were observed between the two groups in terms of sex distribution, mean age, prevalence of comorbidities, or pharmacological treatment. Regarding the cardiological exploratory parameters, there were no differences in mean CHA2DS2-VA score, MVP-ECG risk score, or NT-Pro-BNP levels. However, significant differences were found in the echocardiographic parameters (LA-Sr, 2D LA-EF, and LA-iV), which were significantly more impaired in individuals with AF.

### 3.1. Diagnosis of Atrial Fibrillation

Baseline ECGs were available for all cohort participants. Regarding heart rhythm monitoring with the FibriCheck^®^ App, 149 (100%) of participants completed it at least once, and 120 (80.5%) adhered to the study protocol by performing both monitoring sessions. In 10% of the cases, difficulties were observed in monitoring due to the inadequacy of the mobile model or due to difficulty in handling by the patient, requiring the intervention of the caregiver.

Table 2 presents data on various prognostic variables and their comparison between confirmed and unconfirmed AF diagnoses in the FibriCheck^®^ monitoring cases. A total of 21/149 cases (14.09%) were identified as positive for possible DDAF using FibriCheck^®^. Among these, 12 new cases of AF were diagnosed: 3 cases were detected by baseline ECG, while 9 cases were confirmed during follow-up—3 through Holter monitoring and 6 via ECG. Additionally, one FibriCheck^®^ negative case was diagnosed with atrial flutter by ECG.

Two cases were identified as FibriCheck^®^-positive but not confirmed by ECG and declined Holter monitoring. During the second monitoring, 4/120 new FibriCheck^®^-positive cases (3.3%) were identified among individuals who had tested negative in the previous monitoring. Conversely, 4/14 cases (28.5%) that had been FibriCheck^®^-positive, but without AF confirmed, in the first monitoring tested negative during the second follow-up. The device demonstrated a sensitivity of 57.1% (CI95% 36.1–78.1) and a specificity of 99.2% (CI95% 97.6–100), with FibriCheck^®^-positive individuals having a 71.4-fold higher probability of AF confirmation compared to FibriCheck^®^-negative individuals, resulting in a post-test odds of 87.7%.

APCs were recorded in 13.3% of cases and AHRE in 2.5%. All detected episodes of possible AF lasted longer than 30 s. In 27 cases (18%) there were issues with recording quality and/or with the mobile device’s monitoring capabilities. The average number of self-measurements per patient was similar in those without diagnostic AF compared to those with recorded AF (33.2 ± 19.8 vs. 29.5 ± 16.8, *p* = 0.523), and the percentage of positive measurements was significantly higher in patients who had AF confirmed by Holter (18% vs. 3%, *p* = 0.005).

### 3.2. ECG Variables: MVP ECG Risk Score ≥ 4

The percentage of previously undiagnosed AF detected by basal ECG in high-risk patients was 2.01%. The prevalence of an interatrial block (IAB) was higher in the AF group (53.8% vs. 19.4%, *p* = 0.006). As shown in Table 1, the MVP ECG risk score was higher in those with AF (*p* = 0.003). A total of 16 out of 21 (76.2%) FibriCheck^®^ positives had an MVP ECG score ≥ 4. Of the new AF diagnoses, 11 (84.6%) were made in patients with MVP ECG risk scores ≥ 4. An MVP ECG risk score ≥ 4 increases the risk of AF fourfold [95% CI 2.8–5.2], explains 15.2–18.7% of the dependent variable, and correctly classifies 92.6% of cases. The sensitivity of the MVP ≥ 4 was 84.6%, with a specificity of 56.3%, yielding a 1.9-fold higher probability of confirming AF and a post-test odds of 15.9%.

### 3.3. Echocardiography Study

In total, 140 (93.9%) participants underwent baseline echocardiography, but LA-Sr and/or 2D-LAEF and/or LA volume index were not recorded in 18 cases.

During the baseline evaluation, 79.6% of new AF cases were in the LA-Sr < 26% group (*p* < 0.001). Significant differences were observed between the two groups across all three echocardiographic variables. The prevalence of LA-Sr < 26% and 2D-LAEF < 50% were significantly higher (*p* = 0.002) in new AF cases. The sensitivity of the LA-Sr < 26% was 77.7%, with a specificity of 68.3%, resulting in a 2.4-fold higher probability of confirming AF and a post-test odds of 19.3%.

## 4. Discussion

Results from the PREFATE study indicate that DDAF may be a prevalent finding among individuals at increased risk for AF. The NNS to detect one new case was under 12, which is significantly lower than observed in single-time-point screening studies (NNS = 1/12 vs. 1/147) [31,32]. This highlights the growing recognition of undetected AF, an issue that is critically important given the anticipated 2.5-fold increase in AF prevalence over the next 50 years [1,2]. The lifetime risk of AF is estimated to be 1 in 3-5 individuals over age 55 (37%; 95% CI 34.7–39.6%) [3,4], and AF-related strokes are expected to rise by 34% in the coming decades. Furthermore, these findings shift the focus of discussion from the screening system itself to the decisions made following a probable diagnosis of DDAF or SCAF. This is particularly relevant in primary care settings, where longitudinal monitoring of patients and the identification of high-risk individuals are most feasible.

*Despite the development of* several AF prediction scores, such as CHARGE-AF [33], ARIC AF [34], AFRICAT [35], and others [10], which demonstrate notable predictive performance [36,37], challenges persist in integrating these tools into clinical records for assessing and monitoring patients at elevated risk for AF. Although DDAF episodes are associated with an increased risk of adverse outcomes [38], AF screening has not yet been proven to reduce stroke or systemic embolism or improve overall survival [39]. Consequently, the optimal management of AF detected through screening remains undefined. Table 3 highlights challenges, opportunities, and knowledge gaps in early AF detection in primary care. While healthcare professionals increasingly recognize the importance of AF screening, regulatory bodies have yet to establish clear programs for its implementation [18]. This lack of agreement underscores the need for further research to inform standardized screening protocols.

*Population-based screening for AF could offer several benefits*, including the implementation of comprehensive care strategies to reduce the AF burden and symptoms [8,10], thrombotic risk scoring, and identifying unknown AF cases that could benefit from OAC therapy. Table 1 shows a high percentage of cardiovascular risk factors aligning with variables commonly included in population-based AF risk scores. The EAST-AFNET4 study demonstrated that early rhythm control reduces the composite outcomes of death, stroke, or adverse treatment effects and has revived interest in rhythm control strategies [40]. The reduction in AF burden is already recognized as a therapeutic goal in the ACC/AHA/HRS AF guidelines. Antiarrhythmic drug therapy lowers the average AF burden to less than 3% and successful catheter ablation reduces it to less than 1% [41]. Additionally, early rhythm control therapy reduced AF burden and the incidence of stroke was lower than in usual care (0.6% vs. 0.9%/100 patient-yrs) when the majority of patients were using ACO therapy [42]. This shift in management goals has further heightened interest in identifying DDAF and SCAF cases as critical components of contemporary AF management.

Although irregular rhythms can be detected through pulse palpation, cardiac auscultation, or several devices, an ECG rhythm strip is required for a definitive diagnosis of AF. International guidelines [8,10,24,30] recommend **“*confirmation* via *ECG (12-lead, multiple, or single leads) to establish a clinical AF diagnosis and initiate risk stratification and treatment* (evidence IA)”**. Our results showed that 8 out of 13 new AF cases had prior positive results with FibriCheck^®^ before confirmation by ECG. The ECG *sensitivity was higher than previously reported [14], supporting its use as a screening and follow-up method in populations at high risk for AF alongside the MVP risk score*. AI-based models combining clinical data with ECG analysis have shown AUCs up to 0.90 and may enhance AF screening accuracy, feasibility, and cost-effectiveness [42,43,44,45,46,47]. Not using known AF risk criteria, such as ECG [48,49], echocardiography [21,50], median AF burden [16], or biomarkers [35,43,51,52], may limit the effectiveness of screening methods relying solely on the CHA2DS2-VA score, potentially delaying earlier interventions.

While screening for AF with devices like FibriCheck^®^ has gained attention in recent years, consumer screening faces several significant challenges. False **positives or negatives** may result in either unwarranted reassurance or unnecessary anxiety. In the present study, a lower sensitivity was observed compared to those previously reported [23], which, along with the finding that 28.5% of positive cases in the initial screening turned negative upon subsequent monitoring, highlights the need for confirming AF diagnosis through Holter monitoring. However, this approach may lead to longer wait times for diagnosis and initiation of OAC therapy due to delays in referral cardiology service. The *proposal is to establish a standardized Holter protocol in primary care to facilitate risk assessment and follow-up, thereby reducing waiting times until OAC therapy.* This would help minimize discrepancies in the timing of its application and in the diagnostic criteria for AF, promoting greater consistency in clinical decision-making and improving the quality of care.

In the case of high-risk patients who test negative during monitoring, there is no availability of guidelines regarding the recommended monitoring frequency for AF progression. Several studies [53] have shown that the progression rate from paroxysmal to persistent AF was 8.6% at 1 year, but depending on the co-morbidity profile, progression rates can be as high as 25–35% within the same period. The FibriCheck^®^ device provides a qualitative indicator of the percentage of monitoring with a probable AF result, *but it does not equate to the concept of median AF burden.* Given that in the study new AF cases confirmed by Holter showed a significantly higher percentage of positive measurements with FibriCheck^®^, it suggests that the positive predictive value of the device may increase with a higher percentage of positive measurements. In any case, further *investigation is needed to determine the optimal percentage of positive device measurements that would trigger the implementation of confirmatory Holter monitoring.*

Since AF detected by an external device must be confirmed by ECG recording [54], initiating OAC therapy without ECG confirmation may not align with current guidelines. However, both a DDAF event lasting 5 min and one lasting 3 h may be detected through device monitoring. Yet, it is possible that only the shorter event is confirmed by ECG recording. Consequently, despite identical CHA2DS2-VA scores, OAC therapy would be initiated for the lower AF burden, which is paradoxical considering the high negative and positive predictive values reported for these devices [23]. Evidence from studies such as LOOP, NOAH-AFNET 6, and ARTESiA [55,56,57,58] suggests that even patients with low AF burden are at higher-than-normal stroke risk (~1%/year), the annual stroke risk increases with AF burden, and OAC therapy showed low annual stroke risk in control groups. However, there is uncertainty as to whether DDAF should be treated as clinical AF [59]. Longer SCAF episodes are independently associated with the risk of clinical AF [60].

Device-detected AF and higher CHA2DS2-VA scores at baseline also predict the development of clinical AF [61]. These factors may contribute to the higher stroke risk associated with SCAF episodes as seen in the ASSERT study [62]. The heterogeneity in SCAF definitions and differences in patient populations add complexity in determining the optimal burden [63,64] and duration of SCAF that would benefit from OAC therapy. The threshold of the DDAF/SCAF burden that would trigger OAC initiation is highly variable and will depend on the clinical scenario [65,66]. *The guidelines do not address AF burden, and the CHA2DS2-VA score may not appropriately stratify thromboembolic risk in certain subgroups [8], making it unclear how to categorize the arrhythmia upon first encounter.* Ongoing trials [67] aim to clarify the actual benefits and the risk–benefit ratio of OAC in this specific clinical context, seeking to prevent both overtreatment and excessive bleeding risk.

Bleeding risk assessment has become a crucial paradigm for optimizing the net benefit of antithrombotic treatments in AF patients at high risk of bleeding, such as those in the present study—elderly individuals with polypharmacy, multiple comorbidities, loneliness, and isolation [32,68,69]. The practice of uninterrupted OAC for all AF patients based solely on the CHA2DS2-VA score may be overly simplistic, particularly for those with a low burden of AF or infrequent episodes, potentially leading to overtreatment. A more refined tool *than the HAS-BLED score, which includes labile INR from the VKA era, is needed. Moreover, bleeding risk assessments should be evaluated independently of the AF diagnosis to ensure better-personalized treatment decisions.*

## 5. Study Limitations

We can consider the relatively small sample size, the one-year follow-up period for the progression of DDAF in a condition with an extensive underlying process, and the inherent difficulties related to the availability of mobile devices capable of performing monitoring and handling the FibriCheck^®^ App. Social isolation is another concern for technology adoption by older adults, which may limit accessibility to the benefits of digital technologies. Developing solutions to address this wider societal problem are key to fostering acceptance of digital technologies among older adults [70]. *The results highlight the paradox between the potential uses of technology and the inherent limitations associated with the results, such as treatment decisions and limited accessibility in the context of primary care, which is the most effective setting to tackle a cardiovascular pandemic associated with significant sequelae and costs.*

## 6. Practical Implications and Future Directions

The improved detection and diagnosis of AF, combined with appropriate anticoagulation strategies, will be crucial for improving stroke prevention and reducing its associated social and economic costs [37]. The Action Plan in Europe (2018–2030) prioritizes the availability of detection and treatment programs in primary care to improve the diagnosis and monitoring of populations at risk of AF [1,2]. The integration of all these variables in new IA models [71,72,73] has the potential to provide a platform for patient education about AF and allow personalized treatment decisions in the way of integrated CARE [8,10]. Table 3 presents a set of knowledge gaps as future research directions in early AF detection within primary care.

## 7. Conclusions

The results of this study support the *identification of high-risk individuals for using FibriCheck^®^ for the improved* detection and diagnosis of AF.

The *ECG sensitivity was higher than previously reported*, supporting its use as a screening method in populations at high risk for AF alongside the MVP risk score.

The observation that 28.5% of patients were initially identified as positive for DDAF-FibriCheck^®^ turned negative during subsequent monitoring underscores the importance of Holter monitoring for confirmation.

New AF cases confirmed by Holter had a significantly higher percentage of positive measurements with FibriCheck^®^; the *positive predictive value of the FibriCheck^®^ device may increase with a higher percentage of positive measurements*.

Given the Fibricheck^®^ device does not provide measure of AF burden and the guidelines do not address AF burden, *it remains unclear how to categorize the arrhythmia upon first encounter* when considering the initiation of OAC therapy.

## Figures and Tables

**Figure 1 biomedicines-13-00119-f001:**
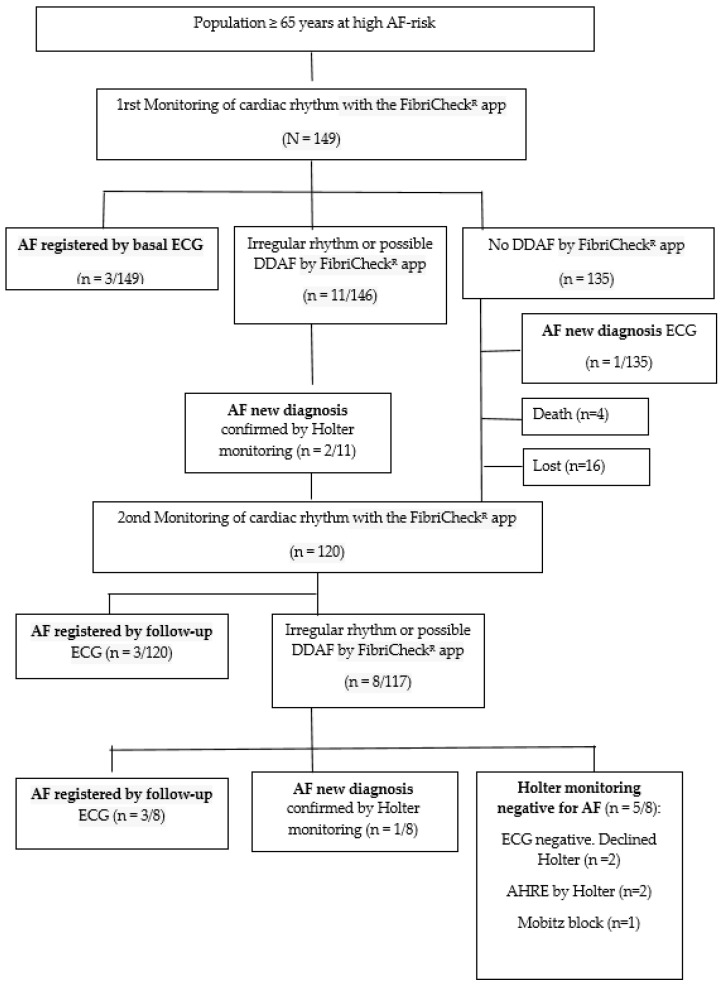
CONSORT (2010) diagram adapted for this study: Screening and follow-up of the study participants. AF: atrial fibrillation; AHREs: atrial high-rate episodes; DDAF: device-detected atrial fibrillation; ECG: electrocardiogram.

**Table 1 biomedicines-13-00119-t001:** Basal characteristics according to Atrial fibrillation vs. no-Atrial fibrillation cases.

Variables	All (%)	AF *	No-AF	*p*-Value *
**N (%)**	149	13 (8.7)	136 (91.2)	
** *General information* **				
Age (years)	74.7 (5.11)	73.46 (6.21)	74.9 (5.0)	0.319
Women	96 (64.4)	6 (6.2)	90 (93.7)	0.224
Men	53 (35.3)	7 (13.2)	46 (86.8)	
BMI (Kg/m^2^)	31.94 (5.50)	33.5 (7.61)	31.8 (4.9)	0.290
** *Comorbidity* **				
Active smoking	13 (8.8)	1 (0.7)	12 (8.1)	0.999
Hypertension	133 (89.9)	13 (100)	120 (80.5)	0.363
Dyslipidaemia	113 (76.4)	9 (69.2)	104 (69.7)	0.507
Diabetes mellitus	76 (51.4)	8 (61.5)	68 (50.0)	0.556
Chronic Renal Failure	36 (24.3)	2 (25.0)	34 (22.3)	0.735
Myocardial ischemia	23 (15.5)	1 (7.7)	22 (14.9)	0.695
Peripheral Vascular disease	13 (6.70)	3 (23.0)	10 (7.3)	0.090
Heart Failure	16 (10.73)	1 (7.7)	15 (11.0)	0.999
Diagnosis of valvular heart disease	9 (6.1)	-	9 (6.1)	0.999
** *Pharmacological treatment* **				
HTA treatment	126 (84.5)	11 (84.6)	115 (84.5)	0.943
Statins treatment	82 (55.03)	7 (53.8)	75 (55.1)	0.999
Diabetes treatment	67 (44.96)	8 (61.5)	59 (43.3)	0.375
Antiplatelet drugs	41 (27.51)	5 (38.4)	36 (26.4)	0.350
** *Cardiological exploratory parameters* **				
CHA_2_DS_2_VA score	3.9 (1.04)	3.9 (0.8)	3.97 (1.0)	0.876
MVP ECG risk score	3.3 (1.4)	4.4 (1.1)	3.2 (1.4)	0.003
Interatrial block (IAB)	33 (22.1)	7 (53.8)	26 (19.1)	0.006
LA-reservoir Strain (%)	28.5 (9.92)	20.4 (13.7)	29.5 (9.1)	0.003
2D-LA-FE (%)	51.6 (12.68)	39.4 (13.4)	52.8 (11.8)	<0.001
LA indexed Vol (mL/m^2^)	30.3 (9.03)	39.0 (9.3)	29.4 (8.7)	<0.001
NT-Pro-BNP	226.2 (300.6)	250.6 (256.5)	217.6 (304.5)	0.77
** *Clinical scores* **				
Pfeiffer score	0.90 (1.2)	1.5 (1.4)	1.0 (1.2)	0.166
Fazecas score	0.84 (0.82)	0.83 (0.8)	0.84 (0.8)	0.965
Fibricheck-measures	32.8 (19.5)	29.5 (16.8)	33.2 (19.8)	0.523

Data are presented as a number of patients (and percentage) or mean (and standard deviation) according to the type of variable. (*) The *p*-value corresponds to the differences in proportions using the Chi-square test for qualitative variables and the *t*-Student test for continuous variables. 2D: two-dimensional; AF: atrial fibrillation; BMI: body mass index; CHA2DS2VA: simplified atrial fibrillation stroke risk score; LA: left atrial; MVP-ECG: electrocardiographic morphology-voltage-P-wave; NT-Pro-BNP: N-terminal pro-brain natriuretic peptide.

**Table 2 biomedicines-13-00119-t002:** Prognostic variables results in FibriCheck^®^ monitoring cases: comparison between confirmed AF vs. no-AF diagnosis.

Case Identifier	CHA2Ds2VA Score	Basal MVP ECG Risk Score	LA-Reservoir Strain (%)	2D-LA-Ejection Fraction (%)	LA-Index Volume (mL/m^2^)	Fibricheck_AF (%) (Number of Measures) − (Rhythm Interpretation) *		Diagnosis Confirmed (ECG or Holter)
						1st Monitoring 2023	2nd Monitoring 2024	
FATE003	4	2/AF	3.7	33.4	53.8	100% [AF]		AF (basal ECG)
FATE019	3	5	34.2	59.5	29.3	0% (11)-[APC]	9.4% (32)-[AF]	AF (Holter)
FATE021	5	5/AF	2.2	10.0	54.1	100% [AF]		AF (basal ECG)
FATE031	5	4	16.4	45.0	28.0	2% (51)-[AF]	0.0% (68)-[APC]	2 * (Holter)
FATE033	5	5	5.0	30.0	32.7	100% [AF]		AF (basal ECG)
FATE050	3	4	-	-	38.0	14.7% (34)-[AF]		AF (Holter)
FATE051	4	3	-	-	-	0% (30)-[APC]	14.3% (7)-[AF]	No confirmed by ECG declined Holter
FATE054	3	3	36.2	54.2	16.8	0% (29)-[SR]	5.0% (20)-[AF]	3 * (Holter)
FATE064	4	5	-	-	-	3.3% (30)-[AF]	0.0% (17)-[SR]	
FATE067	6	6	12.3	32.0	54.9	0% (31)-[SR]	4.3 (23)-[AF]	Flutter (follow-up ECG)
FATE068	4	5	16.6	35.0	30.7	14.3% (35)-[AF]		AF (Holter LA Mixoma)
FATE074	4	5	16.8	56.0	32.0	37.8% (37)-[AF]	23.1% (39)-[AF]	No confirmed by ECG declined Holter
FATE075	3	5	-	-	-	33.2% (205)-[AF]	36.1% (36)-[AF]	3 *(Holter)
FATE076	3	6	24.4	45	32.4	0% (27)-[IEB]	1.4% (29)-[AF]	AF (Holter)
FATE092	5	5	19.4	50.0	36.0	1.4% (38)-[IEB]	52.6% (10)-[AF]	AF (foll0w-up ECG)
FATE094	4	4	34.4	56.0	33.2	0% (22)-[IEB]	63.6% (11)-[AF]	AF (follow-up ECG)
FATE116	3	4	47.0	46.0	29.4	0% (26)-[IEB]		Flutter (basal ECG)
FATE132	3	5/AF	18.4	29.0	43.8	64% (25)-[AF]		AF (follow-up ECG)
FATE133	3	5	-	-	-	3.1% (32)-[AF]		BAV (Holter)
FATE136	4	1	46.8	59.0	35.1	4.3% (23)-[AF]	0.0% (12)-[SR]	
FATE143	5	1	-	26.0	33.5	3.3% (30)-[AF]	0.0% (17)-[APC]	
FATE146	4	4	27.2	47.0	39.7	15.6% (32)-[AF]		AF (follow-up ECG)
All average	3.9 ± 1.04	3.3 ± 1.4	28.5 ± 9.0	51.6 ± 12.6	30.3 ± 9.0			
AF average	3 ± 0.8	4.2 ± 1.1	17.2 ± 8.7	38.6 ± 15.3	38.6 ± 9.3	31.7 ± 11.5		
no-AF average	3.9 ± 1.0	3.2 ± 1.4	28.4 ± 9.1	52.6 ± 11.9	29.6 ± 8.7	43.7 ± 44.2		
*p*-value	0.666	0.017	<0.001	<0.001	0.001	0.022		

* Rhythm interpretation categories: [SR] Sinusal rhythm; [IEB] Isolated ectopic beat; [APC] Atrial premature complexes; [AHRE] Atrial high-rate episodes; [AF] Atrial Fibrillation/Flutter; 2D: two-dimensional; CHA2DS2VA: simplified AF stroke risk score; LA: left atrial; MVP-ECG ECG morphology-voltage-P wave score.

**Table 3 biomedicines-13-00119-t003:** Challenges and opportunities for early AF detection in Primary Care setting.

**1**: Risk stratification for atrial fibrillation should be performed using validated risk scores. Efficient identification of high-risk individuals relies on the routine application of these scores. Additionally, incorporating alerts into clinical records can enhance awareness and facilitate timely intervention.
**2**: The systematic measurement of the MVP score should be included in the risk assessment and documented alongside the CHA2DS2-VA score. Incomplete recording of risk factors and clinical findings in primary care health records can impede the continuity and quality of care.
**3**: Data integration records should be assessed to determine the need for external monitoring, particularly in conjunction with echocardiography findings, such as left atrial ejection fraction (LA-EF) and left atrial strain (LA-Sr). However, limited resources in primary care, including restricted access to echocardiography and external monitoring, pose significant challenges for effective AF screening and follow-up monitoring.
**4**: In cases of a positive result from external monitoring, an atrial fibrillation diagnosis should be confirmed through Holter monitoring, which should be accessible within primary care services. This approach leads to benefits such as empowering users and providers through advanced monitoring, less referral burden, decrease in wait lists, and lower healthcare costs.
**5**: If atrial fibrillation is confirmed, and oral anticoagulation and rhythm control should be initiated in accordance with ESC guidelines. For negative results, a follow-up protocol for external monitoring should be established to ensure ongoing evaluation.
**6**: The availability of quality indicators and cost-effectiveness assessments is essential for evaluating and optimizing the healthcare process. These metrics provide valuable insights into the efficiency, effectiveness, and overall impact of interventions, enabling data-driven improvements in patient care.
**7**: For future research, it is important to emphasize that when atrial fibrillation (AF) (including DDAF and SCAF) is detected via an external device, two additional variables should be assessed to determine whether to initiate oral anticoagulation (OAC): 1/AF burden, in conjunction with other relevant variables, and 2/Thrombotic risk and bleeding risk assessment using artificial intelligence tools that incorporate all of the aforementioned variables independently of the AF diagnosis.

## Data Availability

The data supporting the findings of this study are not currently publicly available but can be requested from the authors upon reasonable request. These data will be available through an institutional repository following the public defense of the corresponding PhD thesis.

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
