# Peer review of "Evidence Gaps and Lessons in the Early Detection of Atrial Fibrillation: A Prospective Study in a Primary Care Setting (PREFATE Study)"

_biomedicines, 2025, doi:10.3390/biomedicines13010119_

Round 1

Reviewer 1 Report

Comments and Suggestions for Authors

The manuscript addresses gaps in the early detection of atrial fibrillation (AF) within primary care, as examined in the PREFATE study, which targets high-risk individuals aged 65–85 years. Employing tools like the FibriCheck® app, ECG, and echocardiography, the study identifies key echocardiographic markers, such as reduced LA strain (<26%), and demonstrates strong predictive value for device-detected AF. While showcasing the viability of systematic screening, it also underscores challenges with current diagnostic criteria and anticoagulation guidelines. The findings highlight the critical role of longitudinal monitoring for DDAF and subclinical AF and advocate for the integration of AI-driven risk assessment and confirmatory Holter monitoring in routine practice. However, the study is constrained by a limited sample size, potential device errors, and challenges in accessibility for older patients.

Clarify the methodology used to randomize participants from electronic health records to improve the reproducibility of the study.

The manuscript indicates a "post-test odds of 87.7%," but the method used to account for pre-test probabilities in this calculation is not sufficiently detailed.

While the FibriCheck® monitoring protocols are well-articulated, the criteria used by the device to label a case as "probable AF" need to be explained in greater depth. Providing a detailed description of the device algorithm would enhance the transparency of the findings.

Additional elaboration is required on the standardization of echocardiographic measurements, particularly for parameters such as LA strain and LA ejection fraction.

The finding that an LA strain below 26% predicts AF is noteworthy. However, it would be valuable to compare these results with established thresholds in the literature for markers of atrial dysfunction to reinforce their validity.

Although the sensitivity (57.1%) and specificity (99.2%) of FibriCheck® are addressed, confidence intervals for these metrics are missing. Furthermore, the 71.4-fold increase in AF probability for device-positive cases requires robust statistical justification to support its significance.

The discussion should place greater emphasis on the practical implications of these results, particularly for clinical decisions regarding anticoagulation therapy and patient follow-up protocols. Additionally, incorporating advanced AI methodologies for patient stratification and analysis could align this work with current trends in digital health. To strengthen the scientific rationale, the authors should also reference key determinants of AF (doi: 10.3390/medicina58111513)

Author Response

The authors sincerely thank the reviewer for the valuable comments and suggestions. The modifications have been incorporated.

Reviewer 1

The manuscript addresses gaps in the early detection of atrial fibrillation (AF) within primary care, as examined in the PREFATE study, which targets high-risk individuals aged 65–85 years. Employing tools like the FibriCheck® app, ECG, and echocardiography, the study identifies key echocardiographic markers, such as reduced LA strain (<26%), and demonstrates strong predictive value for device-detected AF. While showcasing the viability of systematic screening, it also underscores challenges with current diagnostic criteria and anticoagulation guidelines. The findings highlight the critical role of longitudinal monitoring for DDAF and subclinical AF and advocate for the integration of AI-driven risk assessment and confirmatory Holter monitoring in routine practice. However, the study is constrained by a limited sample size, potential device errors, and challenges in accessibility for older patients.

  1. Clarify the methodology used to randomize participants from electronic health records to improve the reproducibility of the study.

(Lines 121-124): The PREFATE project is a prospective observational cohort study (January 1, 2023–December 31, 2024), with its protocol previously published and registered on ClinicalTrials.gov (NCT05772806). The study protocol and previous results have already been detailed in earlier publications [19,20,21]; therefore, they will not be repeated here. It follows the CONSORT (Consolidated Standards of Reporting Trials) Statement (Lines 133-134 and 223-224) and added as Appendix.

  1. The manuscript indicates a "post-test odds of 87.7%," but the method used to account for pre-test probabilities in this calculation is not sufficiently detailed.

(Lines 311-313) The device exhibited a sensitivity of 57.1% (12/21) and a specificity of 99.2%. Individuals with a positive FibriCheck® result had a 71.4-fold increased likelihood of confirmed AF compared to those with a negative result, yielding post-test odds of 87.7%. The methodological approach employed is detailed in:

Argimon-Pallas JM, Jimenez-Villla J. Métodos de Investigación. Aplicados a la atención primaria de salud. Ed Doyma, Barcelona 1993. Anexo 4. Sensibilidad y especificidad, pag: 223-227. ISBN 84-8174-031-4.

Sensitivity (12/21 = 57.1%) Specificity (127/128 = 99.2%) Positive likelihood ratio (PLR) = sensitivity / (100% - specificity): 71.4%. The advantage of using this parameter lies in its independence from the proportion of patients diagnosed with AF in the sample. Instead, it solely depends on the test's sensitivity and specificity, explicitly demonstrating the change between pre- and post-test probabilities. Post-test odds = Pre-test odds (1/10 AF prevalence) x [PLR].

These calculations have been added to the Appendix.

  1. While the FibriCheck® monitoring protocols are well-articulated, the criteria used by the device to label a case as "probable AF" need to be explained in greater depth. Providing a detailed description of the device algorithm would enhance the transparency of the findings.

(Lines 198-200) All cases where the FibriCheck device recorded AF were categorized as "probable" and required confirmation by ECG or Holter monitoring, according to EHRA criteria.

  1. Additional elaboration is required on the standardization of echocardiographic measurements, particularly for parameters such as LA strain and LA ejection fraction.

Indeed, currently, there is no universally accepted standardization for echocardiographic measurements. The measurements utilized in this study were based on the available evidence. The EACVI/EHRA Expert Consensus Document for assessing patients with AF emphasizes the importance of LA strain imaging as a valuable adjunct. Among TTE parameters, increased LA volume and reduced LA ejection fraction (LAEF) have been established as independent predictors of incident AF, stroke, and other adverse cardiovascular events [1]. Recently, two-dimensional speckle tracking echocardiography has emerged as a valuable tool for assessing LA function by quantifying myocardial strain [2]. The reservoir strain, representing the positive systolic peak value of the global LA strain curve during the reservoir phase, has shown the strongest evidence for diagnostic and prognostic utility [2,3] across various clinical populations. It is considered a marker of atrial fibrosis and structural remodelling [4]."

[1] Goette A., Kalman J.M., Aguinaga L., Akar J., Cabrera J.A., Chen S.A., Chugh S.S., Corradi D., D’avila A., Dobrev D., et al. EHRA/HRS/APHRS/SOLAECE expert consensus on atrial cardiomyopathies: Definition, characterization, and clinical implication. Heart Rhythm. 2017;14:e3–e40. doi: 10.1016/j.hrthm.2016.05.028. 

[2] Pathan F., D’Elia N., Nolan M.T., Marwick T.H., Negishi K. Normal Ranges of Left Atrial Strain by Speckle-Tracking Echocardiography: A Systematic Review and Meta-Analysis. J. Am. Soc. Echocardiogr. 2017;30:59–70. doi: 10.1016/j.echo.2016.09.007.

[3] Sade L.E., Keskin S., Can U., Çolak A., Yüce D., Çiftçi O., Özin B., Müderrisoğlu H. Left atrial mechanics for secondary prevention from embolic stroke of undetermined source. Eur. Heart J. Cardiovasc. Imaging. 2020;23:381–391. doi: 10.1093/ehjci/jeaa311. 

[4]. Mannina C., Ito K., Jin Z., Yoshida Y., Matsumoto K., Shames S., Russo C., Elkind M.S.V., Rundek T., Yoshita M., et al. Association of Left Atrial Strain with Ischemic Stroke Risk in Older Adults. JAMA Cardiol. 2023;8:317–325. doi: 10.1001/jamacardio.2022.5449. 

  1. The finding that an LA strain below 26% predicts AF is noteworthy. However, it would be valuable to compare these results with established thresholds in the literature for markers of atrial dysfunction to reinforce their validity.

Indeed, the evidence supports this direction and may be considered a promising avenue for future research. Both a previous article [5] and the preceding discussion provide evidence to corroborate the reviewer's suggestion.

[5]. Gentille-Lorente D, Hernández-Pinilla A, Satue-Gracia E, Muria-Subirats E, Forcadell-Peris MJ, Gentille-Lorente J, Ballesta-Ors J, Martín-Lujan FM, Clua-Espuny JL. Echocardiography and Electrocardiography in Detecting Atrial Cardiomyopathy: A Promising Path to Predicting Cardioembolic Strokes and Atrial Fibrillation. J Clin Med. 2023 Nov 26;12(23):7315. doi: 10.3390/jcm12237315.

Although the sensitivity (57.1%) and specificity (99.2%) of FibriCheck® are addressed, confidence intervals for these metrics are missing. Furthermore, the 71.4-fold increase in AF probability for device-positive cases requires robust statistical justification to support its significance.

They have been added and explained in point 3. Sensitivity 57.1% (CI95% 36.1-78.1). Specificity 99.2% (CI95% 97.6-100)

  1. The discussion should place greater emphasis on the practical implications of these results, particularly for clinical decisions regarding anticoagulation therapy and patient follow-up protocols. Additionally, incorporating advanced AI methodologies for patient stratification and analysis could align this work with current trends in digital health. To strengthen the scientific rationale, the authors should also reference key determinants of AF (doi: 10.3390/medicina58111513)

We fully agree with the reviewer's suggestion. While there is still no consensus on AF screening in high-risk patients, the indication for anticoagulation in these patients requires clearer criteria. Although AI has the potential to reduce this uncertainty, it is not yet included in international clinical practice guidelines. We have incorporated the reviewer's suggestion and the cited article into the original text [Lines 473-475].

Reviewer 2 Report

Comments and Suggestions for Authors

Dear Authors

I have read your paper proposal with great interest. Maybe I am not the best selection for the review since my expertise is mostly associated with experiments, data acquisition and management. I admire your design of an experiment which is well defined and detailed. As an engineer I do not dare to comment on medical stuff but when only numbers and statistics are considered I like your paper very much.

I have some comments

Sentence in lines 81-86 is too long and hard to understand. Can you or maybe a native speaker check the sentences in lines 91 and 97 something does not sound good to me.

I do not understand the tolerance of age in line 209 (is standard deviation or the interval with X% probability)

Definitely a lot of work was done, and a significant amount of data was collected in a very controlled and planed way.

Best regards

Author Response

Reviewer 2

The authors sincerely thank the reviewer for the valuable comments and support regarding the methodology and results of the study. The suggested modifications have been incorporated.

I have read your paper proposal with great interest. Maybe I am not the best selection for the review since my expertise is mostly associated with experiments, data acquisition and management. I admire your design of an experiment which is well defined and detailed. As an engineer I do not dare to comment on medical stuff but when only numbers and statistics are considered I like your paper very much.

I have some comments

  1. Sentence in lines 81-86 is too long and hard to understand. Can you or maybe a native speaker check the sentences in lines 91 and 97 something does not sound good to me.

Done. The paragraph has been rephrased for improved comprehensibility.

  1. I do not understand the tolerance of age in line 209 (is standard deviation or the interval with X% probability)

Fixed. Standard deviation (SD)

Definitely a lot of work was done, and a significant amount of data was collected in a very controlled and planed way.

Best regards

Round 2

Reviewer 1 Report

Comments and Suggestions for Authors

Congratulations to the authors for the quality of their answer.